# Mercury and Antibiotic Resistance Co-Selection in *Bacillus* sp. Isolates from the Almadén Mining District

**DOI:** 10.3390/ijerph18168304

**Published:** 2021-08-05

**Authors:** Marina Robas, Agustín Probanza, Daniel González, Pedro A. Jiménez

**Affiliations:** Department of Pharmaceutical and Health Sciences, Universidad San Pablo CEU, 28668 Boadilla del Monte, Spain; a.probanza@ceu.es (A.P.); daniel.gonzalezreguero@ceu.es (D.G.); pedro.jimenezgomez@ceu.es (P.A.J.)

**Keywords:** *Bacillus*, antibiotic resistance, mercury resistance, resistance co-selection, soil contamination

## Abstract

Antibiotic resistance (AR) in the environment is of great global concern and a threat to public health. Soil bacteria, including Bacillus spp., could act as recipients and reservoirs of AR genes of clinical, livestock, or agricultural origin. These genes can be shared between bacteria, some of which could be potentially human pathogens. This process can be favored in conditions of abiotic stress, such as heavy metal contamination. The Almadén mining district (Ciudad Real, Spain) is one of the environments with the highest mercury (Hg) contamination worldwide. The link between heavy metal contamination and increased AR in environmental bacteria seems clear, due to co-resistance and co-selection phenomena. In the present study, 53 strains were isolated from rhizospheric and bulk soil samples in Almadén. AR was tested using Vitek^®^ 2 and minimum inhibitory concentration (MIC) values were obtained and interpreted based on the criteria of the Clinical and Laboratory Standards Institute (CLSI) guidelines. Based on the resistance profiles, five different antibiotypes were established. The Hg minimum bactericidal concentration (MBC) of each strain was obtained using the plating method with increasing concentrations of HgCl2. A total of 72% of Bacillus spp. showed resistance to two or more commonly used antibiotics. A total of 38 isolates expressed AR to cephalosporins. Finally, the environmental co-selection of AR to cephalosporins and tetracyclines by selective pressure of Hg has been statistically demonstrated.

## 1. Introduction

Antibiotic resistance (AR) of an environmental origin has been attracting the attention of numerous studies in recent years. Its development and evolution in the clinical setting seems to be increasingly evident [1,2,3,4]. Numerous AR genes found in pathogenic bacteria have evolved or have been acquired from environmental microbial communities [5], which suggests the need to study and understand how the environment (soils, waters, sediments, etc.) can be reservoirs of AR. Bacteria that are resistant to antibiotics of clinical importance such as aminoglycosides, β-lactams, glycopeptides, macrolides, quinolones, streptogramin, tetracyclines, and trimethoprim/sulfamethoxazole have been reported [1,3]. Likewise, multidrug-resistant bacteria have also been reported to be common in the environment [6]. For this reason, the World Health Organization (Geneva, Switzerlan) has stated that AR genes are a new pollutant due to their emerging prevalence and extensive distribution [7].

As with AR, it is common to find microorganisms resistant to heavy metals in the natural environment. This resistance is a common phenotype in those microorganisms that have been exposed to metals in their habitats [8].

A good example of an area that is highly contaminated by heavy metals is the Almadén mining district in Ciudad Real, Castilla y León, Spain. The size of the area is approximately 300 km^2^, and it is of geological interest worldwide. Together with the Idrija mine in Slovenia [9], Almadén is considered to have one of the largest deposits of mercury (Hg), with high geogenic levels of Hg as well. The concentration of this heavy metal in the Almadén soil exceeds 106 μg/kg [10]. Mineral Hg deposits are found predominantly as cinnabar (HgS). Elemental Hg (Hg0) is also present and abundant in the local atmosphere, where it can remain for up to 1.7 years [11]. Given Almadén’s large size, as well as the Hg dispersion by rivers and emissions throughout its 2000 years of mining activity, it is considered one of the most contaminated areas on the planet, due to the natural origin of Hg in addition to its anthropogenic exploitation. When the mine closed in 2003, other forms of land use had to be considered, such as agriculture and livestock. However, until economically and environmentally viable strategies are achieved to reduce soil Hg contamination, these productive soil activities cannot be carried out safely.

Hg is a toxic metal, which means that it does not have a specific biological role and its introduction into the body is unnecessary. Therefore, at certain doses Hg produces adverse effects, such as problems with development, growth, and reproduction of living beings [12]. Moreover, this heavy metal has the potential for bioaccumulation and biomagnification, which can be transmitted through the trophic chain, thereby affecting human health [13]. The use of this metal is currently controlled at the European level by Regulation (EU) 2017/852 regarding mercury and, since 2013, following the legally binding UN Convention of Minamata, it has been recognized as a global pollutant [14]. 

The increase in AR genes is considered to be a consequence of the massive use of antibiotics in medical treatment and agriculture [15,16]. However, evidence shows that the spread of AR genes can also be influenced by heavy metal contamination [17,18]. As early as the 1970’s, it was discovered that heavy metal resistance (HMR) and AR can be selected simultaneously in the ecosystem of heavy metal pollutants [19]. At the molecular level, co-selection occurs when two or more genetically linked resistance genes are together, and cross-selection refers to a single genetic element providing tolerance to more than one antimicrobial agent [20]. Among all heavy metals, co-selection of AR and HMR seems most noticeable in the presence of Hg. Resistance genes seem to be highly connected with plasmids, transposons, and integron-associated integrases [21]. This frequent linkage between Hg resistance and AR has been documented in bacteria from a wide range of habitats, including mine sediments [22], freshwater microcosms [23], agriculture soil [24], wastewater treatment systems [25], and water sediment [26]. However, bacterial communities are shaped by a complex series of evolutionary, ecological, and environmental factors [24]. The patterns of HMR and AR genes in an area with long-term contamination by heavy metals have hardly been studied [8], especially in bacterial genera that are predominantly environmental and scarcely associated with a clinical setting, since they do not display pathogenic potential. This is the case of the *Bacillus* genus, which is rarely associated with the appearance of diseases in humans and animals, except for *B. anthracis*, *B. thuringiensis*, *B. sphaericus*, and *B. cereus*.

All members of the *Bacillus* spp. are Gram-positive, endospore-forming bacilli. The latter enhance cell viability, even under conditions of high heavy metal pressure. The *Bacillus* genus is known to produce a wide variety of antimicrobial compounds. This leads to the development of AR genes and different resistance strategies against the drugs that they themselves produce. Furthermore, other environmental bacteria may have developed resistance mechanisms due to their exposure to antibiotics produced by *Bacillus* present in the same environmental niche [25]. With the aforementioned in mind, the environment in general and the soil in particular, seem to be reservoirs of AR genes [27]. These AR genes have the capability of spreading and being transmitted through ecosystems. Moreover, they can be acquired by pathogens that are capable of affecting human health [28]. For this reason, there is growing interest in determining not only the soil antibiotic resistome (comprising all AR genes and their precursors) [29], but also those bacteria that can co-select AR genes in environments with high selective pressure. This bacterial ability for gene co-selection and transmission may pose a risk of dissemination through soil management and ecological processes [28,30] to human pathogens [31,32]. Expanding this knowledge will allow us not only to prioritize the inherent risk to public health, but also to enable the development of appropriate control measures.

The present study aims to analyze the possible co-selection of AR and HMR in the *Bacillus* genus by studying rhizospheric strains isolated from the Almadén mining district, which has been subjected to long-term Hg contamination.

## 2. Materials and Methods

### 2.1. Obtaining Mercury-Tolerant Strains

This study was carried out using rhizospheric and bulk soil samples obtained from an abandoned metallurgical plant in the mining district of Almadén in Ciudad Real, Spain. Experimental “Plot M” is classified as an area with a high level of Hg contamination of up to 1710 mg/kg Hg [33]. The plants sampled were *Rumex induratus* Boiss. & Reut. (A), *Rumex bucephalophorus* L. (B), *Avena sativa* L. (C), *Medicago sativa* L. (D), and *Vicia benghalensis* L. (E). Bulk soil samples were taken from the same plot, in an area free of vegetation. In all cases, the samples were collected in quadruplicate, within the same plot, in order to achieve sufficient mass for this and other assays. For this study, the content of the replicas of each rhizosphere/bulk soil was homogenized until reaching a mass of not less than 2.5 g. For the extraction of bacterial communities, the modified method described by García-Villaraco et al. [34] was employed, suspending 2 g of sample in 20 mL of sterile saline solution (NaCl 0.45%), stirring with blades at 16,000 rpm for 2 min to achieve homogenization. The mixture was then centrifuged at 2500× *g* r.p.m. for 10 min.

The isolation of Hg-resistant bacteria was carried out by mass seeding (1 mL of the supernatant) on Standard Methods Agar plates (SMA, Condalab^®^, Torrejón de Ardoz, Spain) supplemented with 40 µg/mL of HgCl_2_. Bacterial isolates that met Mathema criteria [35] for Gram-positive minimum bactericidal concentration (Hg MBC ≥40 µg/mL) were considered to be resistant to Hg and selected for further analysis.

### 2.2. Molecular Identification of Microorganisms

For the isolation of bacterial DNA, a colony was taken from the pure culture plate grown in SMA and suspended in 100 µL of 1× PBS. The Nucleo spin Gel and PCR clean up^®^ Kit (Machery-Nagel, Strasbourg, France) was used to purify PCR products. A PCR was performed with the MultiGene Mini^®^ kit (Labnet, Madrid, Spain) of a 16S ssrRNA fragment with the primers fD1 5′-GAGTTTGATCCTGGCTCA-3′ and rP2 (5′-ACGGCTACCTTGTTACGACTT-3′) [36]. Ready to go^®^ PCR beads (Healthcare) were used for amplification. For purification of that which was amplified, the Nucleo spin Gel and PCR clean up^®^ Kit was used and the concentration and purity of DNA was measured with DropVue^®^ (General Electric, USA). The sequencing of the PCR products was carried out in the Genomics department at Complutense University of Madrid following the method of Sanger et al. [37] with a 3730xl DNA Analyzer^®^ sequencer (Applied Biosystems, USA). Organisms were identified by comparative bioinformatic analysis with the CLC Sequence Viewer 7 program, and the sequences were aligned with the Nucleotide Massagery tool and EMBL-EBI, Clustal Omega. The consensus sequence (1200 bp) was compared to BLAST from NCBI.

### 2.3. Microbial Diversity

For each of the rhizospheric and bulk soil samples described in 2.1, the microbial diversity was analyzed by calculating the Shannon–Wiener index (*H′*) (bits/ind) [38] according to the following formula:$$H^{\prime }=-\overset{s}{\underset{i=1}{\sum }}\left(\left(\frac{n_{i}}{n}\right)\;\times \;\log_{2}\left(\frac{n_{i}}{n}\right)\right)$$

*H′* = Shannon–Wiener index.

*S* = total number of species.

*n_i_* = number of individuals in a population (species).

*p_i_* = proportional abundance of the *ith* species: $$p_{i}=\frac{n_{i}}{N}$$

*N* = total number of individuals in “*S*” population.

In addition, the Pielou index (*J′*) [39] was calculated to determine the species evenness. *J′* is constrained between 0 and 100 (%) and is calculated using the following formula: $$J^{\prime }=\frac{H^{\prime }}{H_{max}}\;\times \;100$$
where *H_max_* (bits/ ind) is the maximum diversity that would be obtained if the distribution of the abundances of the species in the community were perfectly equitable, and is calculated as follows:$$H_{max}=\log_{2}S$$

### 2.4. Hg Minimum Bactericidal Concentration (MBC)

To study the Hg MBC of the selected bacterial strains (those that necessarily met the MBC mercurotolerance criteria Hg MBC ≥ 40 µg/mL and belonged to the *Bacillus* genus), a Müller Hinton agar (Condalab^®^) was sown in the following HgCl_2_ concentrations: 400, 350, 200, 175, 150, 100, 87.5, 75, 50, and 43.75 µg/mL. The MBC value corresponds to the lowest concentration of HgCl_2_ capable of inhibiting the growth of more than 99.9% of the bacteria.

### 2.5. Antibiogram and Antibiotic Minimum Inhibitory Concentration (MIC)

Inoculum, seeding, and incubation of the AST-ST01 card of the Vitek^®^ 2 system were carried out in accordance with the manufacturer’s specifications (BioMérieux^®^). The data obtained were interpreted according to the guidelines entitled “Performance Standards for Antimicrobial Susceptibility Testing” (CLSI, 2017) [40] and “The European Committee on Antimicrobial Susceptibility Testing. Breakpoint tables for interpretation of MICs and zone diameters” (EUCAST, 2021) [41]. Since there are no generic values for *Bacillus* spp., the values of *Bacillus antrhacis* were used for all strains belonging to this genus.

### 2.6. Statistical Analysis

In order to study the possible correlation between the MIC of the tested antibiotics and the Hg MBC, a linear regression analysis was carried out between the tested antibiotics and the MBC values for each of the studied strains. The statistical program SPSS v.19.0 (SPSS inc.) was used. Significant differences between the values exist when *p*-value ≤ 0.05.

## 3. Results

A total of 149 strains were isolated from bulk soil and the rhizosphere of local plants of the Almadén mining district (“Plot M”). A total of 97 resulted to be Gram positive. Among these, those belonging to the *Bacillus* spp. were selected, making a total of 53 isolates. Figure 1 shows the complete information of the isolated strains, including the origin of isolation and the calculation of the diversity of each sample (rhizosphere of the plant or bulk soil).

Figure 2 shows the results of the antibiogram performed with the automated Vitek^®^ 2 technique, including both MIC breakpoints (Figure 2A) and the interpretation of the breakpoints according to the CLSI guideline criteria (Figure 2B). Bacterial isolates were grouped based on their MICs against different antibiotics into five antibiotypes. Antibiotypes I and Ib are practically the same and include profiles with AR to **¦Â**-lactams and sulfamides. In addition, Ib includes profiles with high MIC to tetracyclines. Antibiotypes II and III correspond to the phenotypes most similar to the wild type. The difference between the two is essentially that antibiotype III includes intermediate AR values for ceftriaxone. Finally, antibiotype IV includes those strains with high profiles of AR to cephalosporins and sulfa drugs.

Figure 3 additionally includes information related to Hg MBC, which ranges in values between 75 µg/mL and 200 µg/mL, being the most frequent mean value 80 µg/mL and 100 µg/mL. In this way, each species with a certain tolerance to Hg (MBC) is related to the antibiotype to which it belongs. Likewise, the frequency (n) with which species appear sharing these characteristics is included. The most common phenotype (n = 11). Is clearly that of *Bacillus toyonensis* with a Hg MBC of 80 µg/mL and belonging to antibiotype I.

The possible correlation between Hg resistance and AR was analyzed using a linear regression (Table 1). Those antibiotics showing significant differences between MIC and MBC variables are indicated with an asterisk (*). In these cases, the MIC of the tested antibiotics covariated with the Hg MBC. Although levofloxacin was tested, the p-value does not appear in the table since all the strains had the same MIC value and was interpreted as a constant in the analysis.

## 4. Discussion

Of the 149 total bacterial strains isolated from the bulk soil and rhizosphere of plants naturally grown in the soil of the Almadén mining district, 65% were identified as Gram-positive bacilli. Of the latter, more than half (54%) were identified as different *Bacillus* spp species. This is something that other authors had already achieved while working with Hg-contaminated soils [42,43], including members of the species *B. thuringiensis*, *B. megaterium*, and others that are less common, such as *B. drentensis*, *B. bataviensis*, and *B. vireti* [44], all of which are also represented in the present study.

Antibiotics are often produced naturally by bacteria, including the genus *Bacillus*, which produces a wide variety of antimicrobial compounds (lipopeptides, enzymes, and antibiotics). At the same time, many of these bacteria have developed genes that are resistant to the antibiotics that they themselves produce. Thus, these antibiotic-producing microorganisms and other bacteria with which they coexist are capable of acting as potential sources of resistance genes [45]. Additionally, environmental bacteria can also be the recipient and reservoir of selected AR genes in the clinical, veterinary, or agricultural production fields. Many authors have demonstrated that resistance to antibiotics of a clinical origin can be traceable to non-pathogenic environmental strains [46,47]. The latter could possibly come into contact with animal and/or human pathogens and end up causing a public health problem [48,49]. 

The interpretation of MIC for environmental isolates is complicated since many of them are non-pathogenic strains. However, MIC breakpoint interpretation for a large group of antibiotics for non-anthracis *Bacillus* spp. can be extrapolated at the genus level using the CLSI M45-P document as a reference [40]. The Vitek^®^ 2 system was used to carry out antibiograms. The automation of this method allows an objective measurement system to obtain the MIC. For the interpretation of MIC values, CLSI does not have disk diffusion criteria for *Bacillus* spp. However, breakpoints are available in the document M100-s16 for *B. anthracis* against penicillin, tetracyclines, and ciprofloxacin [50]. Rare cases of resistance to the two latter antibiotics in *Bacillus* spp. justifies the defining of “susceptible” breakpoints. EUCAST offers criteria for the interpretation of levofloxacin, vancomycin, erythromycin, and clindamycin breakpoints [51]. 

From the set of analyzed strains, 33 belonged to antibiotypes I and Ib and were resistant to both ampicillin and benzylpenicillin. This finding has been widely reported by authors who have suggested that AR genes in numerous species of the *Bacillus* genus are ancient, very widespread, and could be selected by other edaphic species [41]. Less common is the presence of AR to cephalosporins in *Bacillus* spp. strains. However, resistance mechanisms mediated by β-lactamases that confer this phenotype have been described [52]. However, we found that 38 isolates were resistant to cefotaxime and intermediate to ceftriaxone (antibiotypes I, Ib, and IV), which could indicate that AR to cephalosporins in contaminated environments is more common than previously estimated. By contrast, antibiotypes II and III, which hardly present AR phenotypes, were less detected. Notably, wild-type bacteria are in the minority in this type of environment that is heavily polluted by Hg. For this reason, the present study aimed to discover whether the presence of Hg in contaminated soil promotes and fosters the selection of AR mechanisms.

Long-term and persistent heavy metal contamination affects the microbial composition of soils in favor of microbial populations that are better adapted to stress situations, and that are more competitive from an ecological point of view [53]. One way to analyze the impact of this abiotic stress is by studying variations in the community structure and its diversity. The results of the Shannon–Wiener index in this work show a greater diversity in rhizospheric samples than in bulk soils. It is known that plants recruit the bacteria that inhabit the soil around their roots, providing a nutrient-rich microenvironment that favors certain bacterial communities to thrive or aid metal tolerance [54]. Another factor considered for the selection of *Bacillus* spp. strains for the successive analyses was the representativeness between samples. In view of the results and given that the quotient tended toward 100% in the rhizospheric samples, it was concluded that the diversity of the analyzed strains was close to the Hmax expected. This suggests a representativeness of the analyzed samples, eliminating factors that could alter it, such as uneven recruitment of rhizospheric microorganisms by the plant or a different accumulation of Hg in its environment that could modify the distribution of species. It is interesting that in bulk soil, where diversity was lower, 90% of the isolates belonged to “antibiotype I” (Figure 3). Rhizospheric samples, on the other hand, characterized by greater microbial diversity, presented a wider variety of antibiotypes, which could mean that the rhizosphere favors the colonization of bacteria with more heterogeneous AR profiles.

Hg resistance values registered in our strains ranged from 75 to 200 ppm. These concentrations are similar to those described by other authors [45]. It is interesting to see that the highest Hg MBCs are found in the strains of antibiotypes I, Ib, and IV. These antibiotypes are the only ones that are resistant to cephalosporins. The statistical analysis of correlation between Hg MBC and MIC of cefotaxime and ceftriaxone was significant (*p*-value ≤ 0.05). This analysis supports the hypothesis of simultaneous selection of genes for Hg resistance and genes for β-lactamases [19].

The increase in tetracycline resistance is also a serious problem for the clinical treatment of infectious diseases. Therefore, the transmissibility of AR genes is a subject of great scientific interest. Some authors have also observed high levels of AR to tetracyclines in edaphic *Bacillus* spp. strains [55]. Although this phenotype is unusual in the strains of the study herein, the four isolates included in antibiotype Ib showed tetracycline resistance. Traditionally, resistance to tetracyclines in *Bacillus* spp. has been linked to the presence of *tet* genes. These genes are found in a variety of bacteria isolated from humans, animals, and the environment and are typically located on the chromosome, plasmids, and transposons [56]. The mobile nature of these tetracycline-resistance genes may explain their wide distribution both in Gram negatives and Gram positives [57]. Of all the *tet* genes, the *tetL* gene is most commonly found in *Bacillus* plasmids and is the second most prevalent tetracycline-resistant gene in streptococci and enterococci [58]. In their research involving contaminated sediment, Rassmussen and Sørensen [59] found that high levels of AR, at least to tetracycline in environments with high Hg contamination, could be due to the transfer of conjugative plasmids, thus providing indirect evidence of co-resistance. In concordance with these authors, the statistical analysis of the study herein revealed a positive co-selection of resistance to tetracycline and Hg in all of our strains (*p*-value ≤ 0.05).

The interpretation of the significant covariation values between Hg MBC and tetracyclines and cephalosporins MICs agrees with other authors who stated that the HMR and AR genes are co-selected, conferring antibiotic–metal co-resistance [60]. Specifically, the genetic link of Hg resistance and AR in plasmids was demonstrated in bacteria by Summers et al. [61]. The high Hg pressure to which the analyzed soils and strains have been exposed has its origin in mining since Roman times. The findings of the present research coincide with Ji et al. [62], who suggested that metals such as Hg, Cu, and Zn may exert selective pressure for long-term AR genes.

## 5. Conclusions

Most of the mercurotolerant isolates of the *Bacillus* spp. genus are multi-resistant. Only antibiotypes II (*n* = 9) and III (n = 6), poorly represented, presented a resistance profile similar to the wild-type phenotype.Of the 53 strains tested, 38 expressed a phenotype of resistance to cephalosporins. This finding suggests that the incidence of bacterial resistance to these antibiotics in Hg-contaminated soils is higher than estimated in previous works.The correlation between the Hg minimum bactericidal concentration (CMB) and the CMI of cefotaxime, ceftazidime, and tetracycline MIC was significant (*p*-value ≤ 0.05). This finding statistically demonstrates the co-selection of bacterial resistance to cephalosporins and tetracyclines under Hg-selective environmental pressure.

## Figures and Tables

**Figure 1 ijerph-18-08304-f001:**
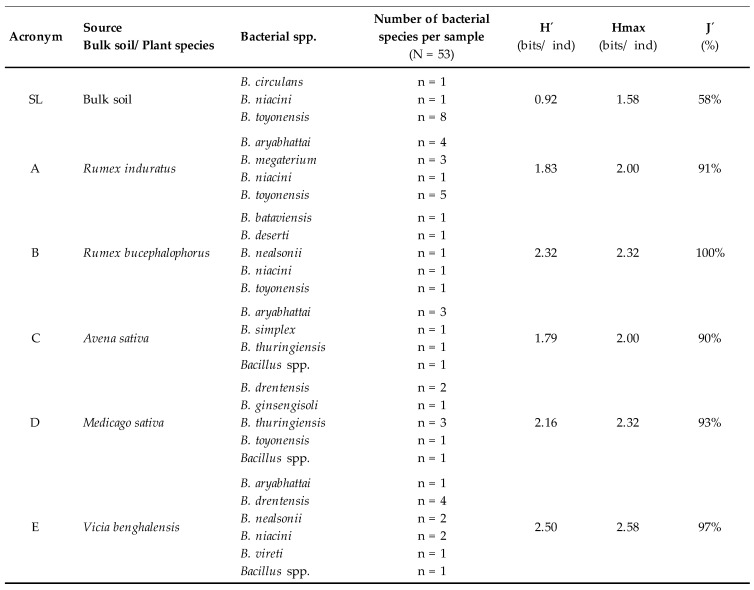
Isolated *Bacillus* spp. strains and diversity of the analyzed samples. H´: diversity calculated using the Shannon–Wiener index (bits/ind), H_max_: maximum diversity (bits/ind), and J´: Pielou index (%).

**Figure 2 ijerph-18-08304-f002:**
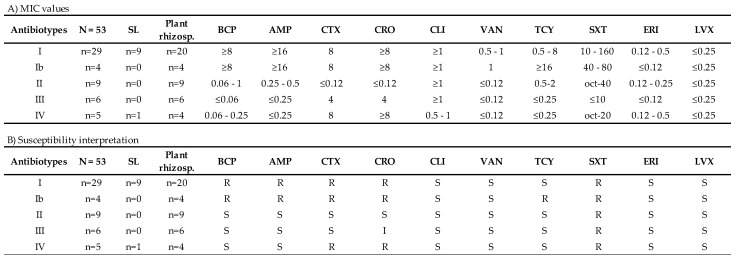
Antibiotic susceptibility of the *Bacillus* strains. SL: *Bacillus* spp. strains isolated from bulk soil. Plant rhizosph.: *Bacillus* spp. strains isolated from assayed plant rhizospheres. Antibiotics included on the Vitek^®^ 2 AST-ST01 card (µg/mL): BCP: benzylpenicillin, CTX: cefotaxime, AMP: ampicillin, CRO: ceftriaxone, LVX: levofloxacin, TCY: tetracycline, ERI: erythromycin, VAN: vancomycin, CLI: clindamycin, LIN: linezolid, SXT: cotrimoxazole. (**A**) MIC values for each of the antibiotics tested against the set of 53 *Bacillus* isolates, and (**B**) breakpoint interpretation according to CLSI and EUCAST guidelines for each tested antibiotic. R: resistant, I: intermediate, S: susceptible.

**Figure 3 ijerph-18-08304-f003:**
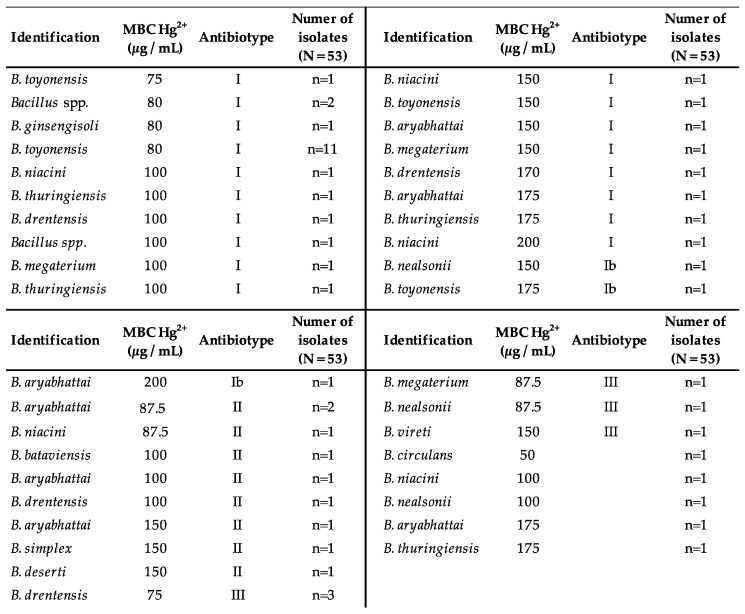
Arrangements between Hg MBC and antibiotypes. n: number of isolates in each situation. MBC: minimum bactericidal concentration.

**Table 1 ijerph-18-08304-t001:** Co-selection of Hg MBC and AR in the isolated *Bacillus* spp. strains.

Family	Antibiotic	*p*-Value
Betalactams	benzylpenicillin	0.597
	ampicillin	0.992
	cefotaxime	0.035 *
	ceftriaxone	0.038 *
Lincosamines	clindamycin	0.051
Glycopeptides	vancomycin	0.988
Tetracyclines	tetracycline	0.001 *
Sulfamides	cotrimoxazole	0.322
Macrolides	erythromycin	0.289
Quinolones	levofloxacin	-

Results of the linear regression between Hg MBC and antibiotic MIC. * Antibiotics showing significant differences (*p*-value ≤ 0.05). Levofloxacin does not have a p-value because the MIC for all strains was constant, without variation with the independent variable.

## Data Availability

No new data were created or analyzed in this study. Data sharing is not applicable to this article.

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
