# Peer review of "Mercury and Antibiotic Resistance Co-Selection in *Bacillus* sp. Isolates from the Almadén Mining District"

_ijerph, 2021, doi:10.3390/ijerph18168304_

Round 1

Reviewer 1 Report

Robas et al. examined Bacillus recovered from plants and soil from a metallurgical plant to study the co-selection of Hg resistance and antibiotic resistance. The paper does not read well and it is not ready for publication yet. In fact, the manuscript does not contain ample description of the results and the implications of the findings. Most of the discussion section needs to be deleted; the authors provided background information and described the results of other researches, making it sound like a literature review. Discussion section must contain the meaning of the results the authors obtained, implications of the findings, possible improvements to further develop the topic, etc. Also, the authors need to revise the manuscript for English language as there are many grammar errors, odd vocabulary, and long complicated sentences, making sentences difficult to understand (I have pointed out few of them below).

Here are some specific comments:-

Authors have introduced the abbreviation AR but uses “antibiotic resistance” and “antimicrobial resistance” throughout the manuscript. Also, both ARG and AR genes were used. Please be consistent

Lines 25-29: The first paragraph of the introduction section seems a bit plain. Maybe add a sentence or two on why AR is an emerging global problem. Also, add references for the second and third sentences of the first paragraph.

Line 32: Almaden Mining District in (specific city or country)?

Lines 33-34: avoid short sentences; explain why Hg is a serious environmental threat

Line 36: “sediments and waters” and “16,000 ug/g and...”

Line 36: 11,2 ug/L? perhaps 11.2 ug/L or 11,2000 ug/L?

Line 37: revise “to the order of 14.0 ug/m3

Line 43: use other words such as, eliminate, instead of “withstand”?

Lines 46-48: provide the references

Line 50: Hg

Lines 53-57: How are the two sentences contrary to each other? Is it genomes (first sentence) or plasmids (second sentence) that make the difference, or the environment affected by abiotic pressure and the environment not affected by those pressure make the difference in the presence of metal resistance genes (if so, mention it in the first sentence)?

Line 57: The authors have not discussed about their results yet, so this sentence needs to be modified accordingly or be moved to the discussion section

Line 59 (several other places as well): resistance genes    

Line 61: does not need “horizontally”

Lines 61-62: genes that confer resistance to

Line 65: the sentence that ends with “... that transport” sounds incomplete 

Line 66: italicize “mer”. Also, introduce what mer operon is

Line 67: resistance operons

Line 68: introduce the abbreviation ARG

Line 70: delete “also” since the previous sentences weren’t talking about the rapid evolution of Hg resistance

Lines 71-74: subjects are not clear in this sentence. What is “contributing to the appearance of resistance” and what is “it” in line 73? Specify. Also, “evolution” instead of “appearance” and “plasmid-mediated resistance” instead of “plasmid resistance”?

Line 75: isolated from where?

Line76: introduce the abbreviation PGPR

Line 76: what do you mean by “give them further use”?

Lines 82-88: This whole sentence needs to be revised. Special attention to the parallelism and the numbering system (insert the number at right places)

Line 86: such as

Lines 89-94: This paragraph is important as this contains the objectives of the research but the sentences do not read well. Please make them clearer and simpler.

Line 99: For the materials, please mention the name of the manufacturer and the place of manufacturing

Line 102: This study was carried out

Line 102: specify what samples i.e. soil and plant

Line 109: briefly describe for the plants as well, indicating the modifications

Line 113: which agar plates? Provide the references for the methods as well

Line 119: How about “For the isolation of bacterial DNA, a colony was taken from each of the pure culture plate and suspended...”

Lines 125-126: how about “Nucleo spin Gel and PCR clean up® Kit was used to purify PCR products”

Line 135: remove Hg for the correct abbreviation

Line 136: selected bacterial strains. Also, how did you make the selection?

Line 142: AST-ST01 gallery or card?

Line 144: guideline

Line 149: “statistical analysis” instead of “information processing”

The first and second paragraphs under the Result section can be combined

Line 158: belonged

Line 159: revise ”Three were identified at the gender level”- specify what “three” indicates (three isolates?), do you mean “genus level”?

Line 159: “all of them”- specify what all of them are

Lines 162-163: Didn’t you already mention that these isolates are Gram positive? Just say “53 Bacillus isolates”

Line 162: The authors presented the results in Table 1 but do not further describe the results, without which the Table 1 is difficult to understand. The authors should either provide the breakpoints for resistance so that the readers know which of the isolates are resistant to which of the antibiotics, and/or provide the summary of the table, mentioning important findings (e.g. resistance to tetracycline was found in XX% of the isolates, resistance to Hg was seen in only XX species)

Table 1: “species” instead of “Identification 16S rRNA”

Lines 175-179: The whole paragraph can be deleted as this is methods rather than results. Or this paragraph could be reduced to just a simple sentence, such as “The possible relationship between resistance to Hg and resistance to different families of antibiotics was analyzed using a linear regression” as an opening sentence for the next paragraph

180-184: why blank?

Line 201: high pressure heavy metal conditions?

Line 205: a large part or most?

Line 223: the environment and non-pathogenic strains

Lines 228-230: revise the sentence and also provide references

Author Response

Dear reviewer:

First of all, we want to thank you for your time and dedication. Your comments have been of great value to improve the manuscript. For this reason, we have incorporated all your suggestions.

As a result, you will see that the document has undergone numerous modifications. Especially in the introduction, results and discussion section. There are also somewhat minor changes, in materials and methods and in conclusions. We agree with you that adequate data processing was lacking. They were exposed very rough and the reader was not helped in their interpretation.

Here are the main changes:

1. The introduction has been simplified and has accommodated discussion paragraphs that were improperly placed.

2. Materials and methods: the analysis of diversity has been incorporated, through the calculation of the Shannon-Wiener and Pielou indices. The objective is to justify the suitability of searching for strains in the rhizosphere of plants, where diversity is greater, and to verify the representativeness between samples.

3. Results: we have suppressed Table 1 (MIC values) and replaced it with two explanatory tables (Table 2, where the data from the old Table 1 are grouped into 5 antibiotypes and interpretation values ​​of the breakpoints and Table 3, where Hg MBC values ​​are assigned to antibiotypes). In addition, a new Table 1 has been included with the origin of the isolation of the strains and the diversity calculations.

4. Discussion: It has been practically totally reformulated. We have included a more in-depth analysis of our results explaining the findings. We have moved all the introductory paragraphs. Likewise, its extension has been reduced.

5. Conclusions: they have been reformulated, in order to make them more concrete.

6. Bibliography: as a consequence of the above, there may be variations (additions and deletions) of some of the citations.

We apologize for not leaving the changes in "change control". We did it, but there were too many of them and their follow-up was very complicated. Likewise, our document has been reviewed by a native Philologist, in order to minimize possible language errors.

We hope that this new version meets the expectations of the "Major Revision".

Thanks again for your time.

Marina

***

Reviewers considerations:

  • Authors have introduced the abbreviation AR but uses “antibiotic resistance” and “antimicrobial resistance” throughout the manuscript. Also, both ARG and AR genes were used. Please be consistent -- Corrected.
  • Lines 25-29: The first paragraph of the introduction section seems a bit plain. Maybe add a sentence or two on why AR is an emerging global problem -- corrected. Also, add references for the second and third sentences of the first paragraph -- changed.
  • Line 32: Almaden Mining District in (specific city or country)? -- corrected.
  • Lines 33-34: avoid short sentences; explain why Hg is a serious environmental threat -- included.
  • Line 36: “sediments and waters” and “16,000 ug/g and...” -- eliminated.
  • Line 36: 11,2 ug/L? perhaps 11.2 ug/L or 11,2000 ug/L? -- eliminated.
  • Line 37: revise “to the order of 14.0 ug/m3” -- eliminated.
  • Line 43: use other words such as, eliminate, instead of “withstand”? -- corrected.
  • Lines 46-48: provide the references -- eliminated.
  • Line 50: Hg -- corrected.
  • Lines 53-57: How are the two sentences contrary to each other? Is it genomes (first sentence) or plasmids (second sentence) that make the difference, or the environment affected by abiotic pressure and the environment not affected by those pressure make the difference in the presence of metal resistance genes (if so, mention it in the first sentence)? -- eliminated.
  • Line 57: The authors have not discussed about their results yet, so this sentence needs to be modified accordingly or be moved to the discussion section -- eliminated.
  • Line 59 (several other places as well): resistance genes -- corrected.
  • Line 61: does not need “horizontally” -- reformulated
  • Lines 61-62: genes that confer resistance to -- corrected.
  • Line 65: the sentence that ends with “... that transport” sounds incomplete -- eliminated.
  • Line 66: italicize “mer”. -- corrected.
  • Line 67: resistance operons -- eliminated.
  • Line 68: introduce the abbreviation ARG -- eliminated.
  • Line 70: delete “also” since the previous sentences weren’t talking about the rapid evolution of Hg resistance -- eliminated.
  • Lines 71-74: subjects are not clear in this sentence. What is “contributing to the appearance of resistance” and what is “it” in line 73? Specify. Also, “evolution” instead of “appearance” and “plasmid-mediated resistance” instead of “plasmid resistance”? -- reformulated
  • Line 75: isolated from where? -- reformulated
  • Line76: introduce the abbreviation PGPR -- eliminated.
  • Line 76: what do you mean by “give them further use”? -- eliminated.
  • Lines 82-88: This whole sentence needs to be revised. Special attention to the parallelism and the numbering system (insert the number at right places). -- reformulated
  • Line 86: such as -- reformulated
  • Lines 89-94: This paragraph is important as this contains the objectives of the research but the sentences do not read well. Please make them clearer and simpler. -- reformulated
  • Line 99: For the materials, please mention the name of the manufacturer and the place of manufacturing
  • Line 102: This study was carried out -- corrected.
  • Line 102: specify what samples i.e. soil and plant -- corrected.
  • Line 109: briefly describe for the plants as well, indicating the modifications
  • Line 113: which agar plates? Provide the references for the methods as well
  • Line 119: How about “For the isolation of bacterial DNA, a colony was taken from each of the pure culture plate and suspended...” -- corrected.
  • Lines 125-126: how about “Nucleo spin Gel and PCR clean up® Kit was used to purify PCR products”. -- corrected.
  • Line 135: remove Hg for the correct abbreviation -- corrected.
  • Line 136: selected bacterial strains. Also, how did you make the selection? -- corrected.
  • Line 142: AST-ST01 gallery or card? -- corrected (card).
  • Line 144: guideline -- corrected.
  • Line 149: “statistical analysis” instead of “information processing” -- corrected.
  • The first and second paragraphs under the Result section can be combined -- reformulated.
  • Line 158: belonged -- reformulated.
  • Line 159: revise ”Three were identified at the gender level”- specify what “three” indicates (three isolates?), do you mean “genus level”? -- reformulated.
  • Line 159: “all of them”- specify what all of them are -- reformulated.
  • Lines 162-163: Didn’t you already mention that these isolates are Gram positive? Just say “53 Bacillus isolates” -- reformulated.
  • Line 162: The authors presented the results in Table 1 but do not further describe the results, without which the Table 1 is difficult to understand. The authors should either provide the breakpoints for resistance so that the readers know which of the isolates are resistant to which of the antibiotics, and/or provide the summary of the table, mentioning important findings (e.g. resistance to tetracycline was found in XX% of the isolates, resistance to Hg was seen in only XX species) -- reformulated (New: Tables 2 and 3).
  • Table 1: “species” instead of “Identification 16S rRNA” -- corrected.
  • Lines 175-179: The whole paragraph can be deleted as this is methods rather than results. Or this paragraph could be reduced to just a simple sentence, such as “The possible relationship between resistance to Hg and resistance to different families of antibiotics was analyzed using a linear regression” as an opening sentence for the next paragraph. -- corrected.
  • 180-184: why blank? -- corrected.
  • Line 201: high pressure heavy metal conditions? -- reformulated.
  • Line 205: a large part or most? -- reformulated.
  • Line 223: the environment and non-pathogenic strains -- reformulated.
  • Lines 228-230: revise the sentence and also provide references -- reformulated.

Reviewer 2 Report

This article uses the Almadén mining area in Spain, which is heavily polluted by mercury, as the sample collection site. Several strains of bacteria have been isolated from the rhizosphere bacteria of oats, alfalfa and other plants that grow in the mining area, and genetic technology has been used to prove that these bacteria can be tolerated High concentrations of mercury have low drug resistance and can be used as biological agents to repair soils in mercury-contaminated areas. A total of 149 strains of bacteria were isolated in this paper, 97 of which were Gram-positive bacteria. The large number of bacteria isolated made the conclusion more persuasive. However, the analysis of mercury tolerance and drug resistance of Bacillus in these Gram-positive bacteria only seemed to be biased. If other categories of bacteria can be isolated by the same analysis of mercury tolerance and drug resistance, the experimental content will be more complete. In addition, the experimental part of this paper is too few, with only two charts and thousands of words. Too much space is devoted to discussing the research results of others, which makes it seem like reading a review rather than a research result. If the isolated bacteria can be used in practice, For example, it will be inoculated in mercury-polluted soil and measured the change of mercury content in the soil and the tolerance of bacteria to antibiotics, which will enrich the content of the article and make the experimental results more convincing.

  1. In line 28 of the article, why does it say that metals can promote antibiotic resistance? Can you explain it in detail?
  2. In lines 30 and 31 of the article, why it is said that metals resistant to toxins are very ancient as well can be obtained that genes resistant to toxins are very ancient as well. Can you explain it in detail?
  3. Lines 45 and 46 of the article say that pollutants are transformed into soil, water and air. Does it mean that pollutants are directly transformed into soil, water and air, or does the form of pollutants change? The language of the article is inaccurate, please modify it carefully.
  4. A series of biological information comparison operations are carried out in the article. Why do we need to perform this operation? How can this experiment operation help you explain the purpose of your experiment? Why is the comparison result not displayed or analyzed in the article.
  5. The discussion part of the article is too long and it is not conducive to reading. Whether part of the background explanation part of the article is moved to the introduction.
  6. The experimental operation in this article is relatively simple. Can the selected strain be verified?

Author Response

Dear reviewer:

First of all, we want to thank you for your time and dedication. Your comments have been of great value to improve the manuscript. For this reason, we have incorporated all your suggestions.

As a result, you will see that the document has undergone numerous modifications. Especially in the introduction, results and discussion section. There are also somewhat minor changes, in materials and methods and in conclusions. We agree with you that adequate data processing was lacking. They were exposed very rough and the reader was not helped in their interpretation.

Here are the main changes:

1. The introduction has been simplified and has accommodated discussion paragraphs that were improperly placed.

2. Materials and methods: the analysis of diversity has been incorporated, through the calculation of the Shannon-Wiener and Pielou indices. The objective is to justify the suitability of searching for strains in the rhizosphere of plants, where diversity is greater, and to verify the representativeness between samples.

3. Results: we have suppressed Table 1 (MIC values) and replaced it with two explanatory tables (Table 2, where the data from the old Table 1 are grouped into 5 antibiotypes and interpretation values ​​of the breakpoints and Table 3, where Hg MBC values ​​are assigned to antibiotypes). In addition, a new Table 1 has been included with the origin of the isolation of the strains and the diversity calculations.

4. Discussion: It has been practically totally reformulated. We have included a more in-depth analysis of our results explaining the findings. We have moved all the introductory paragraphs. Likewise, its extension has been reduced.

5. Conclusions: they have been reformulated, in order to make them more concrete.

6. Bibliography: as a consequence of the above, there may be variations (additions and deletions) of some of the citations.

We apologize for not leaving the changes in "change control". We did it, but there were too many of them and their follow-up was very complicated. Likewise, our document has been reviewed by a native Philologist, in order to minimize possible language errors.

We hope that this new version meets the expectations of the "Major Revision".

Thanks again for your time.

Marina

***

This article uses the Almadén mining area in Spain, which is heavily polluted by mercury, as the sample collection site. Several strains of bacteria have been isolated from the rhizosphere bacteria of oats, alfalfa and other plants that grow in the mining area, and genetic technology has been used to prove that these bacteria can be tolerated High concentrations of mercury have low drug resistance and can be used as biological agents to repair soils in mercury-contaminated areas. A total of 149 strains of bacteria were isolated in this paper, 97 of which were Gram-positive bacteria. The large number of bacteria isolated made the conclusion more persuasive. However, the analysis of mercury tolerance and drug resistance of Bacillus in these Gram-positive bacteria only seemed to be biased. If other categories of bacteria can be isolated by the same analysis of mercury tolerance and drug resistance, the experimental content will be more complete. In addition, the experimental part of this paper is too few, with only two charts and thousands of words. Too much space is devoted to discussing the research results of others, which makes it seem like reading a review rather than a research result. If the isolated bacteria can be used in practice, For example, it will be inoculated in mercury-polluted soil and measured the change of mercury content in the soil and the tolerance of bacteria to antibiotics, which will enrich the content of the article and make the experimental results more convincing

Answer: We appreciate your input. This study is focused on knowing if the phenomenon co-selection to Hg and antibiotics occurs in the genus Bacillus. In other genera, the phenomenon (compared to other heavy metals) is better described, especially in Gram negative ones. There is very little described about the genus Bacillus because in itself it is not a human pathogen, however, it is a reservoir of these genes, especially if they are co-selected in this type of environment. We believed that in itself, work with this genre would be novel as well as interesting, due to the potential clinical dynamics of these findings. In view of the results and the existence of co-selection, more specific tests may be carried out to specifically check the mechanisms behind this fact. In addition, few groups work with Hg and there are better models for other less toxic heavy metals (Cd, Cr, Zn, ...).

The problem with working with Streptomyces spp., Brevibacterium spp., Paenebacillus spp., Which are the Gram-positive genera that we have also isolated, is that to date there are no references (CLSI, EUCAST) that allow the interpretation of the results. It will be something that will be addressed later, when we continue to increase the "N" of each bacterial genus and we can do MIC tests that allow us to draw conclusions.

Regarding the traceability of Hg in soils, in a trial that is part of a larger study on plant growth-promoting bacteria. Although we have promising results (we have seen phenotypically and genotypically a phyto-protection by these Hg-tolerant strains, through the overexpression of merA), they will be the subject of another set of trials and publications.

  1. In line 28 of the article, why does it say that metals can promote antibiotic resistance? Can you explain it in detail? -- reformulated.
  2. In lines 30 and 31 of the article, why it is said that metals resistant to toxins are very ancient as well can be obtained that genes resistant to toxins are very ancient as well. Can you explain it in detail? -- reformulated.
  3. Lines 45 and 46 of the article say that pollutants are transformed into soil, water and air. Does it mean that pollutants are directly transformed into soil, water and air, or does the form of pollutants change? The language of the article is inaccurate, please modify it carefully. -- reformulated.
  4. A series of biological information comparison operations are carried out in the article. Why do we need to perform this operation? How can this experiment operation help you explain the purpose of your experiment? Why is the comparison result not displayed or analyzed in the article -- Sorry, we have not come to understand this contribution. If it had not been corrected, we will be pleased to proceed, if you give us more indications.
  5. The discussion part of the article is too long and it is not conducive to reading. Whether part of the background explanation part of the article is moved to the introduction. -- reformulated.
  6. The experimental operation in this article is relatively simple. Can the selected strain be verified? – reformulated and remediated.

Round 2

Reviewer 1 Report

Based on the amount of revisions made, in my opinion, the manuscript should have been submitted as a new submission. Also, this revised version is difficult to read as there are too many arrows and lines everywhere. I am sure there is a format for the revised manuscript, so please use that.

General comments:-

  1. What was the reason for collecting bulk soil samples? How did that help achieve your goal of investigating co-selection of AR and HMR? After all, the aim of the study included only “rhizospheric strains” (line 379).

Have you noticed any significant difference in AR and MBC between strains obtained from rhizosphere and bulk soil?

Why did you examine bacterial diversity? Did you investigate bacterial diversity even though you knew you would find greater diversity in rhizosphere and chose to investigate rhizosphere because of that reason (line 1032)?

Explanations for the above should be included the Introduction section when you introduce the aim of the study.

  1. For the materials, please mention the name of the manufacturer and the place of manufacturing
  2. Throughout the manuscript, please check the font style and size, especially tables
  3. Revise the manuscript for English language, not only for grammatical errors but also for smooth flow of contents, especially in the Introduction and Discussion sections. For the introduction, paragraphs need to be re-arranged and transitional sentences/words need to be used for smoother transitions. Also, try to delete unnecessary information and combine paragraphs.
  4. Check the line numbers. There are missing lines.
  5. The first two paragraphs under the Discussion section contain good information but they are not suitable for Discussion section as they are. They could be moved to Introduction section. Otherwise, combine them with your results to discuss the results of your research and see if the results correspond to your initial hypothesis

Specific comments:-

Line 52: italicize Bacillus (check throughout the manuscript)

Line 53: “genes” instead of “mechanisms”

Line 54: delete “Likewise,”

Line 58: delete: “thanks to” and replace with something else

Line 58: 53 bacterial strains

Line 59: soil samples in Almaden

Line 59: delete “using AST-ST01 (Vitek2®-BioMerièux)” as this information is too detailed for an abstract

Line 60: un-capitalize Minimum Inhibitory Concentration

Line 61: introduce the abbreviation CLSI

Line 63: HgCl2- subscript 2

Line 76: “bacteria” instead of “phenotypes”?

Lines 77-78: ... clinical importance, such as aminoglycosides, ... sulfamethoxazole, have been reported

Lines 76-78: provide the references

Line 79: ... common in the environment

Line 191: on the planet

Line 192: When did the mine close? Were the other forms of land use just “considered” or actually carried out?

Line 201: specify “it”

Line 214: system

Line 222: It is a new paragraph and usually a new paragraph contains a new topic, so make sure you specify what “this genus” is.

Lines 222-223: it sounds like “which enhance cell viability, even under conditions of high heavy metal pressure” is endospore-forming bacilli when it is endospores that match the description

Line 234: specify “This”

Lines 227-228: antibiotics produced by Bacillus present in the same environmental niche, including soil

Line 229: AR genes instead of antibiotic resistance genes

Line 233: AR genes instead of ARGs

Line 384: how many samples were collected? Please specify for both bulk soil samples and rhizosphere samples

Line 387: a high level of mercury contamination

Line 387: at or above?

Line 398: un-capitalize Maximum Bactericidal Concentration

Line 560: Hg Maximum Bactericidal Concentration. Also, isn’t this Minimum Bactericidal Concentration, especially when it is the “lowest concentration” (line 563)? Make the change throughout the manuscript

Line 561: How did you “select” the bacterial strains? Why not all and how did you make the selection?

Line 576: between the MIC of the tested antibiotics

Line 581: from rhizosphere as well as bulk soil, or only from rhizosphere?

Table 1: It is a confusing table

- rewrite the table title by deleting “origin”

- Change the headings to: “source (bulk soil/plant species)” and “number of isolates (total number =53)”

- there is information on the number of isolates obtained per sample but the information on the number of each bacterial species is absent

- under H’, fix the decimal (period instead of comma)

- “strain number” column can be deleted

Line 658: including both MIC breakpoints...

Line 660: delete “the”

Line 661: what do you mean by “coincident” here?

Line 662: what do you mean by “high AR”? high MIC or high frequency?

Line 670: “53 Bacillus isolates” instead of “53 Gram positive strains of the Bacillus genus”

Table 2:

- this table needs a title and lines 667-672 need to be in the footnotes

- what do R, I, and S stand for?

Line 688: It is not Table 2. It is Table 2.

Line 688: Hg MBC

Lines 690-692: abbreviations need to be at the footnotes

Line 688 and 690: Table 2 is actually Table 3. I see that the isolates are “arranged” but do not see any “grouping”. Most importantly, I do not see why this table would be necessary.

Line 720: Table 3 is actually Table 4

Line 722: un-capitalize the drug names

Lines 731-735: Are these results from other studies? They sound like the results from this study. Revise the sentences. Also, the Discussion should discuss your results in comparison to other results, but this paragraph only discusses other research studies.

Lines 991: delete “There is growing concern about the spread of resistance genes in the environment. Line 992: delete “Likewise”

Line 1007: According to these sentences, you have used both CLSI and EUCAST breakpoints, but the method section (and also line 671) only mentioned CLSI breakpoints and not EUCAST breakpoints

Line 1009: 33 isolates belonged

Please check the Result section for tense (use past tense except for few cases) e.g. belonged (line 1009), were resistant (line 1015), aimed (line 1020)

Line 1012 (less common... AR to cephalosporins) contradicts line 1014 (38 isolates are resistant); therefore, instead of “In this sense”, it should be “However, we found that 38 isolates were resistant to cefotaxime and intermediate to ceftriaxone (antibiotypes I, Ib, and IV)”

Line 1017: delete “until now”

Line 1018: “found less common” or “least detected” instead of “in the minority”

Line 1019: did you mean “wildtype”?

Line 1019: maybe “paradoxical” is not a right word. It is widely known that the presence of heavy metals promotes antimicrobial resistance

Lines 1023-1025: these two sentences can be deleted

Line 1025: chronic contamination of what? Specify

Line 1026: populations that are better adapted

Line 1027: what is “it”? specify

Lines 1029-1030: reference needed

Lines 1023-1153: this paragraph is overall verbose and very confusing. Try to reduce explanations on why you have done you did, and discuss the results you obtained and what those results mean.

Lines 1156-1158: specify “it” and “these genes”. Also, reference needed

Lines 1154-1163: This paragraph may be deleted as this study does not investigate mer operons or any mobile genetic elements

Please specify for clarification: this phenomenon (line 1173), this phenotype (line 1174), its presence (line 1175)

Line 1179, 1181: resistance gene

Line 1793: introduce the abbreviation CMB

Reference 49: M100-S16 is not from 2020

Reference number 49 should be 51

Author Response

General comments:-

The answers to several questions are addressed in the next answer:

  1. What was the reason for collecting bulk soil samples? How did that help achieve your goal of investigating co-selection of AR and HMR? After all, the aim of the study included only “rhizospheric strains” (line 379).
  2. Have you noticed any significant difference in AR and MBC between strains obtained from rhizosphere and bulk soil?
  3. Why did you examine bacterial diversity? Did you investigate bacterial diversity even though you knew you would find greater diversity in rhizosphere and chose to investigate rhizosphere because of that reason (line 1032)?

Answer: The objective when the study was planned was to have a collection of strains of the Bacillus genus large enough (N> 30) for the statistics to be conclusive. Finally, we obtained a total of 53 strains that met the selection criteria: Bacilli, Gram positive, aerobic, endospore forming, mercurotolerant (CMB > 40ppm). All of them were identified at the genus level as Bacillus spp. It means, therefore, that we discarded many other Gram positive rods that did not meet the rest of the criteria (Clostridium spp., Streptomyces spp., Brevibacterium spp., Actinomyces spp.). The reason for focusing on Bacillus and not on the rest is because although they are not clinically active, they do have some species that are (B. cereus, B. anthracis, especially). This guarantees that there are reference values ​​for the interpretation of MICs at the gender level.

When sampling, we were looking for representativeness, that is, to ensure that the study was representative at the genus level and not at the species level. Therefore, the more diversity the better. Based on the criteria of the bibliography, the greatest diversity is at the rhizospheric level, which is why we resorted to sampling 5 plant species. We include bulk soil to corroborate this information and to avoid the bias of the "plant effect" if it existed.

Finally, we obtained 10 individuals from the bulk soil and we wanted to count on them, in order to carry out a cross-sectional study of the Bacillus in the soil. Being so few (N <10) the statistics were complicated, but nevertheless, we found out: 1) there is no significant variation between the Hg at the rhizospheric level and the free soil level (the variation of the mean of the MBC in rhizosphere and soil free is not significantly different) and 2) There are no significant differences in the behavior of the strains taking the origin of the sample as an independent variable.

Explanations for the above should be included the Introduction section when you introduce the aim of the study.

  1. For the materials, please mention the name of the manufacturer and the place of manufacturing -- incorporated
  2. Throughout the manuscript, please check the font style and size, especially tables – verified.
  3. Revise the manuscript for English language, not only for grammatical errors but also for smooth flow of contents, especially in the Introduction and Discussion sections. For the introduction, paragraphs need to be re-arranged and transitional sentences/words need to be used for smoother transitions. Also, try to delete unnecessary information and combine paragraphs -- Thanks for the suggestion. In order to satisfy this requirement, it has been reviewed (grammatically and syntactically) by a native English philologist.
  4. Check the line numbers. There are missing lines – We regret that this continues to happen. For some reason, the footer counts it and there is a jump in the numbering in a systematic way. We have tried to fix it from several computers and word processors without success. Please excuse us in advance.
  5. The first two paragraphs under the Discussion section contain good information but they are not suitable for Discussion section as they are. They could be moved to Introduction section. Otherwise, combine them with your results to discuss the results of your research and see if the results correspond to your initial hypothesis -- Analysis of our findings has been reformulated and incorporated.

Specific comments:-

Line 52: italicize Bacillus (check throughout the manuscript) -- corrected

Line 53: “genes” instead of “mechanisms” -- corrected

Line 54: delete “Likewise,” -- corrected

Line 58: delete: “thanks to” and replace with something else – corrected and replaced by “due to”

Line 58: 53 bacterial strains -- corrected

Line 59: soil samples in Almaden -- corrected

Line 59: delete “using AST-ST01 (Vitek2®-BioMerièux)” as this information is too detailed for an abstract -- deleted

Line 60: un-capitalize Minimum Inhibitory Concentration – corrected throughout the manuscript.

Line 61: introduce the abbreviation CLSI – incorporated.

Line 63: HgCl2- subscript 2 -- corrected

Line 76: “bacteria” instead of “phenotypes”? -- corrected

Lines 77-78: ... clinical importance, such as aminoglycosides, ... sulfamethoxazole, have been reported -- corrected

Lines 76-78: provide the references – [1, 3]

Line 79: ... common in the environment -- corrected

Line 191: on the planet -- corrected

Line 192: When did the mine close? Were the other forms of land use just “considered” or actually carried out? – corrected. The mine was closed in 2003. A paragraph has been added clarifying that it is a need, which is being worked on and is not being carried out on a regional scale.

Line 201: specify “it” -- corrected (Hg)

Line 214: system -- corrected

Line 222: It is a new paragraph and usually a new paragraph contains a new topic, so make sure you specify what “this genus” is. – corrected (Bacillus spp.)

Lines 222-223: it sounds like “which enhance cell viability, even under conditions of high heavy metal pressure” is endospore-forming bacilli when it is endospores that match the description – corrected and clarified.

Line 234: specify “This” – corrected “the bacterial ability for gene co-selection and transmission.

Lines 227-228: antibiotics produced by Bacillus present in the same environmental niche, including soil”

Line 229: AR genes instead of antibiotic resistance genes -- corrected

Line 233: AR genes instead of ARGs -- corrected

Line 384: how many samples were collected? Please specify for both bulk soil samples and rhizosphere samples – included.

Line 387: a high level of mercury contamination -- corrected

Line 387: at or above? – corrected “up to”

Line 398: un-capitalize Maximum Bactericidal Concentration -- corrected

Line 560: Hg Maximum Bactericidal Concentration. Also, isn’t this Minimum Bactericidal Concentration, especially when it is the “lowest concentration” (line 563)? Make the change throughout the manuscript -- corrected throughout the manuscript

Line 561: How did you “select” the bacterial strains? Why not all and how did you make the selection? This sampling was carried out not only for the realization of this test, but also for a set of analyzes. Altogether we isolated more than 150 bacteria belonging to different genus and species. In this project we focus on the selection and study of the co-selection of Hg and AR in Bacillus. In parallel, the same analysis has been carried out in Gram negatives (Pseudomonas and Xanthomonas). Finally, 11 strains could not (initially) be identified and we are working on them. Therefore, the selection criteria for the present studies: Gram positive, aerobic (excluding Clostridium spp, for example), spore-forming and mercurotolerant (MBC> 40ppm).

Line 576: between the MIC of the tested antibiotics – corrected

Line 581: from rhizosphere as well as bulk soil, or only from rhizosphere? – corrected (also bulk soil)

Table 1: It is a confusing table -- corrected

- rewrite the table title by deleting “origin”

- Change the headings to: “source (bulk soil/plant species)” and “number of isolates (total number =53)”

- there is information on the number of isolates obtained per sample but the information on the number of each bacterial species is absent

- under H’, fix the decimal (period instead of comma)

- “strain number” column can be deleted

Line 658: including both MIC breakpoints... -- corrected

Line 660: delete “the” -- corrected

Line 661: what do you mean by “coincident” here?  -- corrected “practically the same”

Line 662: what do you mean by “high AR”? high MIC or high frequency? – corrected “high MIC”

Line 670: “53 Bacillus isolates” instead of “53 Gram positive strains of the Bacillus genus” -- corrected

Table 2: -- corrected

- this table needs a title and lines 667-672 need to be in the footnotes

- what do R, I, and S stand for?

Line 688: It is not Table 2. It is Table 2. -- corrected

Line 688: Hg MBC -- corrected

Lines 690-692: abbreviations need to be at the footnotes -- corrected

Line 688 and 690: Table 2 is actually Table 3. I see that the isolates are “arranged” but do not see any “grouping”. Most importantly, I do not see why this table would be necessary. – corrected. We have considered eliminating it, however, we would lose the information regarding CMB Hg. For this reason and because it allows us to link it with the frequency of antibiotypes, we have chosen to leave it.

Line 720: Table 3 is actually Table 4 -- corrected

Line 722: un-capitalize the drug names -- corrected

Lines 731-735: Are these results from other studies? They sound like the results from this study. Revise the sentences. Also, the Discussion should discuss your results in comparison to other results, but this paragraph only discusses other research studies – reformulated.

Lines 991: delete “There is growing concern about the spread of resistance genes in the environment. Line 992: delete “Likewise” -- corrected

Line 1007: According to these sentences, you have used both CLSI and EUCAST breakpoints, but the method section (and also line 671) only mentioned CLSI breakpoints and not EUCAST breakpoints – corrected and included in methods.

Line 1009: 33 isolates belonged -- corrected

Please check the Result section for tense (use past tense except for few cases) e.g. belonged (line 1009), were resistant (line 1015), aimed (line 1020) – corrected and checked.

Line 1012 (less common... AR to cephalosporins) contradicts line 1014 (38 isolates are resistant); therefore, instead of “In this sense”, it should be “However, we found that 38 isolates were resistant to cefotaxime and intermediate to ceftriaxone (antibiotypes I, Ib, and IV)” -- corrected

Line 1017: delete “until now” -- corrected

Line 1018: “found less common” or “least detected” instead of “in the minority” -- corrected

Line 1019: did you mean “wildtype”? Yes, -- corrected

Line 1019: maybe “paradoxical” is not a right word. It is widely known that the presence of heavy metals promotes antimicrobial resistance – corrected and replaced by “notably”

Lines 1023-1025: these two sentences can be deleted – eliminated.

Line 1025: chronic contamination of what? Specify – corrected and replaced by “long-term and persistent”.

Line 1026: populations that are better adapted -- corrected

Line 1027: what is “it”? specify – corrected “the impact of abiotic stress”.

Lines 1029-1030: reference needed – corrected [55]

Lines 1023-1153: this paragraph is overall verbose and very confusing. Try to reduce explanations on why you have done you did, and discuss the results you obtained and what those results mean. – corrected and reformulated and included our results.

Lines 1156-1158: specify “it” and “these genes”. Also, reference needed – corrected “operon”, “Hg resistance” and [56]

Lines 1154-1163: This paragraph may be deleted as this study does not investigate mer operons or any mobile genetic elements – deleted.

Please specify for clarification: this phenomenon (line 1173), this phenotype (line 1174), its presence (line 1175) – corrected “AR to tetracyclines”; 0; Tetracycline resistance.

Line 1179, 1181: resistance gene -- corrected

Line 1793: introduce the abbreviation CMB -- corrected

Reference 49: M100-S16 is not from 2020 -- corrected

Reference number 49 should be 51 -- corrected

Reviewer 2 Report

There are some format problems in this paper, such as lines688-692 and Conclusions.

Author Response

Thank you very much for your evaluation. We have made an exhaustive review of the format throughout the document, paying special attention to the sections that you mention.

Likewise, we take the opportunity to regret that the problem of line numbering persists. For some reason, the footer counts it and there is a jump in the numbering in a systematic way. We have tried to fix it from several computers and word processors without success. Please excuse us in advance.
